# Inter-Comparison of Radon Measurements from a Commercial Beta-Attenuation Monitor and ANSTO Dual Flow Loop Monitor

Matthew L. Riley [1,]*[ ], Scott D. Chambers [2][ ] and Alastair G. Williams [2][ ]

1 New South Wales Department of Planning and Environment, Sydney 2141, Australia
2 Australian Nuclear Science and Technology Organisation, Sydney 2234, Australia; szc@ansto.gov.au (S.D.C.); agw@ansto.gov.au (A.G.W.)
* Correspondence: matthew.riley@environment.nsw.gov.au

**Abstract:** Radon (Rn) is a radioactive, colourless, odourless, noble gas that decays rapidly. It's most stable isotope, $^{222}$Rn, has a half-life of around 3.8 days. Atmospheric radon measurements play an important role in understanding our atmospheric environments. Naturally occurring radon can be used as an atmospheric tracer for airmass tracking, to assist in modelling boundary layer development, and is important for understanding background radiation levels and personal exposure to natural radiation. The daughter products from radon decay also play an important role when measuring fine particle pollution using beta-attenuation monitors (BAM). Beta radiation from the $^{222}$Rn decay chain interferes with BAM measurements of fine particles; thus, some BAMs incorporate radon measurements into their sampling systems. BAMs are ubiquitous in air quality monitoring networks globally and present a hitherto unexplored source of dense, continuous radon measurements. In this paper, we compare in situ real world $^{222}$Rn measurements from a high quality ANSTO dual flow loop, dual filter radon detector, and the radon measurements made by a commercial BAM instrument (Thermo 5014i). We find strong correlations between systems for hourly measurements ($R^2 = 0.91$), daily means ($R^2 = 0.95$), hour of day ($R^2 = 0.72$–$0.94$), and by month ($R^2 = 0.83$–$0.94$). The BAM underestimates radon by 22–39%; however, the linear response of the BAM measurements implies that they could be corrected to reflect the ANSTO standard measurements. Regardless, the radon measurements from BAMs could be used with correction to estimate local mixed layer development. Though only a 12-month study at a single location, our results suggest that radon measurements from BAMs can complement more robust measurements from standard monitors, augment radon measurements across broad regions of the world, and provide useful information for studies using radon as a tracer, particularly for boundary layer development and airmass identification.

**Keywords:** radon; BAM; beta attenuation; air quality monitoring; monitoring networks



## 1. Introduction

Radon ($^{222}$Rn) is a radioactive noble gas that is a component of the uranium decay chain. It has a half-life ($t_{1/2}$) of 3.82 days and is the immediate product of the decay of radium ($^{226}$Ra, $t_{1/2} = 1600$ y). Radium occurs naturally in rocks and soils and is ubiquitous across land globally. Hence, radon is emitted continuously by land masses, although the magnitude of this emissions flux varies dependent on soil properties [1].

Pioneering German scientists Julius Elster and Hans Geitel first discovered radioactive elements in the air in 1901. In 1904, they identified that their source was "radium emanation" from the soil [2]. Initially, radon measurements were primarily undertaken to assess the human health impacts of naturally occurring radiation, particularly in houses and underground (such as mines) [3]. However, since the 1990s there has been a growing use of radon as an atmospheric tracer [4–6] for global climate model (GCM) [7–9] and chemical transport model evaluation [10], in studies of boundary layer meteorology [11–14], for urban

climate and air pollution studies [15–18], and in identifying and quantifying greenhouse gas emission sources [19–21].

Radon has direct impacts on human health. It is an identified human carcinogen and can induce gene mutations and chromosomal aberrations [22]. Exposure to radon is the second leading cause of lung cancer [23]. Recent work also suggests that radon and its progeny, attached to particulates, can act as "significant effect modifier of PM2.5-associated total, cardiovascular, and respiratory mortality" and that radon may enhance PM2.5-associated mortality [24].

Continuous radon measurements are conducted globally by numerous monitoring and research networks. These include the World Meteorological Organisation's Global Atmosphere Watch [25], the Integrated Carbon Observation System [26], and the United States Environmental Protection Agency's RadNet [27], among others. However, the distribution of radon measurements is heterogenous, there are gaps in the global coverage, and there is a lack of harmonisation among the various measurement techniques [28]

Currently there are three principal methods used for continuous radon observations: (1) direct measurement through dual flow-loop, twin filter detectors [29], (2) measurement of the radioactive decay of radon progeny attached to particles and collected on a single filter paper [30], and (3) electrostatic deposition [31,32]. The dual flow-loop, twin filter system of the Australian Nuclear Science and Technology Organisation (ANSTO), provides direct, high-precision measurements with low minimum detection limits [33–35]. The ANSTO system is the recommended instrument in GAW and ICOS networks [25,26].

There have been several studies that compare radon measurement techniques. Xia et al., (2010) [36] compared the ANSTO system to a single filter monitor over the course of one year at a mountain top site in south-western Germany. They found that the monitors followed the same patterns and gave correlations of $R^2$ between 0.68 and 0.90. Schmithüsen et al., (2017) [37] undertook a European wide comparison of the dual and single filter systems by co-locating portable HRM instruments with the resident ANSTO systems at three locations. In the same study, they also compared different single filter instruments at six locations. They find that the different systems are sufficiently comparable to support simple linear corrections between the systems. Recently, Grossi et. al. (2020) [38] compared the three main techniques for short periods (2 months, 3 weeks) at two sites southwest of Paris. They found correlations ($R^2$) between the ANSTO and HRM instruments of 0.90–0.93.

The single filter method requires assumptions about the radioactive disequilibrium between $^{222}$Rn and its measured progeny in the atmosphere, which changes with height above ground and is largest near the surface [37]. Furthermore, aerosol removal processes such as dry or wet deposition (including rain and fog) may bias the measurements [36], and the $\alpha$-activity of long-lived decay products of ambient thoron ($^{220}$Rn) may accumulate on filters and require separation via spectroscopy [30].

In contrast to instruments designed specifically for radon measurement, beta-attenuation monitors (BAM) are designed to measure atmospheric aerosols. BAMs estimate particle mass by measuring the attenuation of beta radiation (usually from a $^{14}$C source) by solid particles deposited onto a filter [39]. The measurement of beta-attenuation due to particles can be affected by naturally occurring beta sources, primarily radon progeny. Hence, some BAM instruments include estimates of atmospheric radon concentrations in their measurement system, using approaches similar to the single filter radon systems.

Unlike radon instruments, BAMs are far more widespread due to their common application to measuring particles within air quality monitoring networks. In Europe alone there are many hundreds of BAMs used in air quality monitoring networks reporting to the European Environment Agency (https://discomap.eea.europa.eu/App/AQViewer/index.html?fqn=Airquality_Dissem.b2g.Measurements) (accessed on 1 August 2023).

Our motivation for this study was driven by two ideas. If measurements from the two instruments are similar under most meteorological conditions, then BAMs operating routinely in air quality monitoring networks could provide useful radon data to supplement the radon measurements undertaken in other monitoring networks, filling gaps in data

coverage. If their responses correlate and are linear, then simple correction factors can be applied to the BAM radon data.

Further, if the instrument responses do not significantly vary by time, season, or prevalent weather conditions then radon measurements from BAMs could be used to characterise site specific boundary layer development. This outcome may be of particular interest to boundary layer, urban, and air pollution meteorologists, who are all interested in the diurnal and seasonal development of the mixed layer at local scales. This is not a new concept. Perrino et al., (2001) utilised radon progeny measurements from a commercially available BAM (SM200, OPSIS AB, Furulund, Sweden) to characterise atmospheric stability in Rome, Italy, over a 12-month period [40]. The SM200 has also been marketed as a "stability monitor" and has been recently used to study atmospheric stability in Lanzhou and Jinhua, China [41,42]. However, outside of Italy and China there have been few studies using the radon measurements of BAMs and none that we are aware of that use the Thermo Fisher Scientific family of BAMS.

We are also unaware of any studies that compare radon measurements from BAMs to any of the commonly used radon specific instruments, in either laboratory of field-based evaluations. Here we compare radon measurements from a commercially available BAM to those from the ANSTO system.

## 2. Materials and Methods

### 2.1. Site Locations

Measurements were conducted at Liverpool, a suburb of Sydney, the most populous city in Australia (pop. 5.3 million). Sydney, the capital of the state of New South Wales (NSW), is a mid-latitude (34° S, 151° E) coastal basin city bounded to the east by the Pacific Ocean, to the west by the world heritage Blue Mountains, which rise to an altitude of 1189 m.a.s.l., to the north by the Hawkesbury River and to the south by the Woronora Plateau and Georges River. Sydney has a humid subtropical climate (Köppen–Geiger, Cfa) with warm and hot summers and cool winters.

The geology of the Sydney Basin is dominated by Triassic shales and sandstone. The sand that was to become the sandstone of today was washed there by rivers from the south and northwest and laid down between 200 and 360 million years ago. Sydney features two major soil types: sandy soils (which originate from the Hawkesbury sandstone) and clay (which are from shales and volcanic rocks) [43].

Liverpool (pop. 27,000) is located approximately 25 km west-southwest of the Sydney CBD (Figure 1a). It is the major commercial centre of southwest Sydney and supports diverse commercial, light-industrial, service, healthcare, and education industries. The surrounding residential regions are predominantly suburban tract housing with some areas of higher density low-rise apartments.

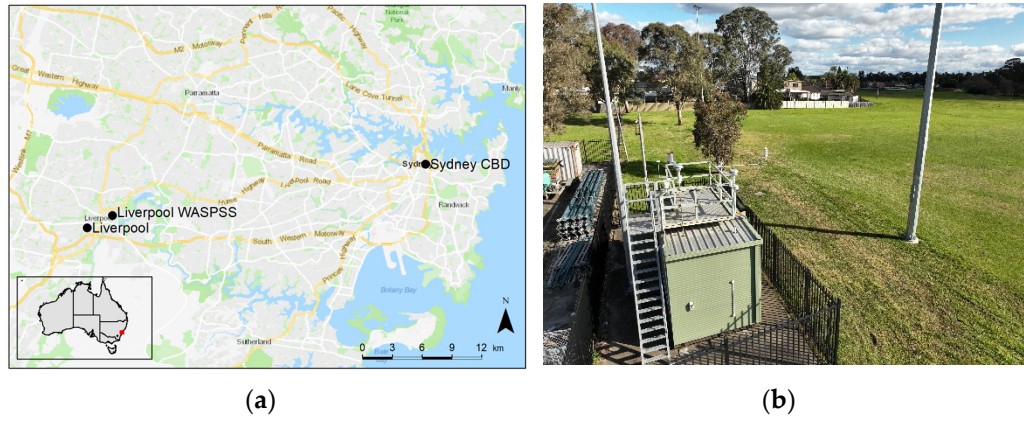

(**a**)                                                              (**b**)

**Figure 1.** Study location showing orientation to the Sydney CBD and site separation (**a**) and the Liverpool DPE station (**b**).

The NSW Government Department of Planning and Environment (DPE) has a long-term air quality monitoring station located in a compound at Rose St (33.93° S, 151.91° E) [44] (Figure 1b). The site monitors local meteorology, a range of pollutants. including $O_3$, $NO_2$, NO, NOx, $SO_2$, and black carbon, visibility, PM10, and PM2.5. PM2.5 is measured using a beta attenuation monitor (see below).

As part of the Western Air-Shed and Particulate Study for Sydney (WASPSS, (https://nespurban.edu.au/research-projects/air-quality/) (accessed on 1 August 2023), a second monitoring station was temporarily located at Liverpool Girls High School (33.92° S, 150.93° E) approximately 2.6 km northeast of the DPE station. The station monitored a similar suite of pollutants but with PM2.5 monitored by tapered element oscillating microbalance (TEOM). ANSTO operated their dual loop twin filter radon monitor at the site.

### 2.2. Study Period

The DPE station has operated continuously since 1990. The temporary monitoring station at the high school operated from early 2019 through to 2021. We focus on results from the 12-month period of 16 March 2019 to 15 March 2020.

This period coincided with a strong El Nino event, with record temperatures and well below average rainfall, contributing to unprecedented fires and smoke events across south-eastern Australia [45]. The El Nino broke down in mid-2019 before moving towards neutral conditions for the remainder of the study period. The period ended with above average rainfall in February [46].

### 2.3. Instrumentation

#### 2.3.1. ANSTO Radon Detector

The ANSTO dual-flow-loop two-filter radon detector provides direct measurement of radon concentrations ensuring that observations are not influenced significantly by measurement height, precipitation, fog, mixing conditions, or aerosol loading [2,28,33]. This detector typically yields detection limits an order of magnitude lower that of other commonly available radon measurement techniques [6]. The precision and sensitivity of the ANSTO measurement system has been acknowledged by its inclusion as a standard radon measurement instrument in the World Meteorological Organisation (WMO) Global Atmosphere Watch program. It is also the most widely used monitor in the European radon network [47].

A detailed description of a comparable sampling system is provided in [33] and the principal of operation is explained in detail in [34,48,49]. In short, sampled air is first stored for 5–6 min to remove the short-lived gaseous radioisotope thoron ($^{220}$Rn; t1/2 = 55.6 s). The sample is then filtered to remove ambient radon and thoron progeny (particulates) and passed into a large delay volume (1500 L). Full volumetric exchange in the delay chamber occurs every 20 min, during which time new radon progeny form in an otherwise aerosol-free environment. An internal flow loop (the second flow loop) operates at approximately 4–5 times the sampling flow rate, collecting and measuring newly formed radon progeny on a second filter before they decay. Detectors are calibrated monthly and instrumental maintenance and background checks performed quarterly.

#### 2.3.2. BAM 5014i

The DPE station operates a 5014i BAM Continuous Ambient Particulate Monitor (Thermo Fisher Scientific Inc.) in order to sample PM2.5. The inlet is at a height approximately 4 m above ground level. The sample is drawn at a volumetric flow rate of 16.7 L/min through a Very Sharp Cut Cyclone (VSCC$^{TM}$) head. The instrument is housed in a small monitoring shed that is temperature controlled (Figure 1b).

The beta attenuation technique of measuring particle mass relies on measuring the attenuation of beta radiation (in our case from a $^{14}$C source < 3.7 MBq) by solid particles deposited onto filter media. The amount of beta radiation attenuated by the particles is exponentially dependent on the particle mass alone and not on other features (such as density, chemical composition, optical properties, etc.) [48].

However, the $^{14}$C source is not the only source of beta radiation that the sample system is exposed to. Naturally occurring $^{222}$Rn and its daughter nuclides can also attach to airborne particles. When these particles are collected on the BAM sample filter they continue to decompose. Beta radiation released during the decay phase, notably of Pb $\rightarrow$ Bi, which can interfere with the BAM measurements (Figure 2).

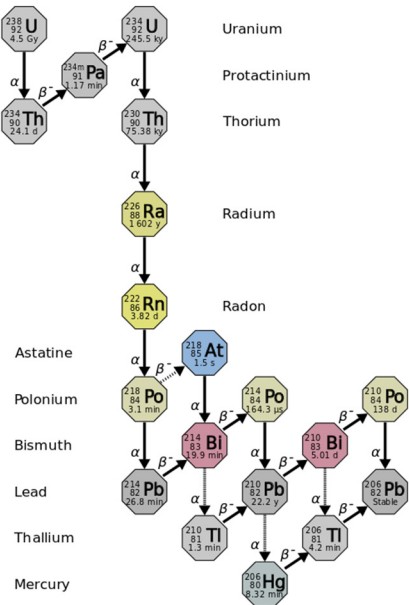

**Figure 2.** Uranium-238 decay chain. Source: Tosaka (unaltered), https://commons.wikimedia.org/wiki/File:Decay_chain(4n%2B2,_Uranium_series).svg. (accessed on 1 August 2023) (Creative Commons CC BY 3.0).

To control for these interferences from naturally occurring beta radiation, the BAM incorporates a radon measurement estimate based on response from a proportional detector (LND4335, LND Inc., New York, USA) that measures $\alpha$ and $\beta$ particles. It estimates radon based on variations in the proportion of both $\alpha$ and $\beta$ particles collected on a filter tape positioned between the $^{14}$C source and the detector. The $\alpha$ particles emitted as $^{222}$Rn $\rightarrow$ $^{218}$Po $\rightarrow$ $^{214}$Pb can be related to the $\beta$ particles released during subsequent decay chain $^{214}$Pb $\rightarrow$ $^{214}$Bi $\rightarrow$ $^{214}$Po.

From the manufacturer, the activity concentration ($C_{Rn}$) of $^{222}$Rn is calculated as:

$$C_{Rn} = \left( \frac{1}{\varepsilon_{\alpha2}} \right) \left( \frac{\alpha_n - \alpha_0}{QT_{222}} \right) \tag{1}$$

where:

$\varepsilon_{\alpha2}$ = detection efficiency of $\alpha$ particles
$\alpha_n$ = gross count rate [s$^{-1}$]
$\alpha_0$ = background $\alpha$ count rate with an unloaded filter [s$^{-1}$]
$Q$ = air flow rate [m$^3$ s$^{-1}$]
$T_{222}$ = 4550 s; an equilibrium constant for $^{222}$Rn daughter nuclides

The instrument applies Equation (1) once a radiological equilibrium of $^{222}$Rn decay is reached. The manufacturer reports this as approximately 90 min after a filter change. During that period, the $C_{Rn}$ value is calculated immediately before the filter change is used in order to correct particle measurements. We note that the manufacturer's stated time to equilibrium (90 min, 5400 s) differs from the equilibrium constant $T_{222}$ in (1), which is assigned a value of 4550 s. Even considering a 300 s smoothing function applied to the detectors data capture, we cannot explain the 850 s variance in reported radiological equilibrium.

The filter tape collects particles for a designated period before advancing (every 8 h in our setup). The filter changes are controlled by the instrument that halts the pump, lowers the vacuum chamber plate, advances the filter tape a fixed length, raises the vacuum chamber plate, and initiates the pump operation and a zeroing of the sample spot. Immediately after a filter change, a new measurement cycle is initiated with an automatic zero adjustment of the mass signal. Automatic filter changes also occur if the mass on the filter exceeds 1500 μg or if the flow rate through the filter tape is reduced by more than 5% due to potentially restrictive particle deposition.

Flow, temperature, humidity, and pressure sensors are audited/calibrated at least quarterly. The proportional $\alpha/\beta$ detector is calibrated at least annually using a known mass source on a calibration foil.

### 2.3.3. Meteorological Measurements

Both stations operated standard DPE meteorological measurement systems [44]. This included 10 m horizontal wind measured by sonic anemometer (MetOne 50.5, MetOne Inc., Grants Pass, OR, USA) and temperature and humidity via Vaisala HMP155 (Vaisala Oyj, Finland). The HMP145 probe is housed in a non-aspirated radiation shield at a height approx. 2.5 m above ground level. Each instrument houses internal pressure sensors used for volumetric corrections. For this comparison, we use only the meteorological data from the DPE Liverpool station as this station more closely meets exposure requirements for meteorological measurements.

### 2.4. Data Handling and Analysis

We begin by compiling hourly radon measurements (8784 h) from the two systems and removing all data that failed the relevant instrument quality control checks. As expected, data recovery from the ANSTO sensor was very high (8577 h, 97.9%) with data loss exclusively due to scheduled calibration and maintenance checks.

In contrast, data recovery from the BAM was much lower (7359 h, 84%). The lower data recovery is due to two main factors. First, during the Black Summer bushfire event, communications with the instrument were lost on several days (31 December 2019–2 January 2020, 4 January 2020–6 January 2020, 12 January 2020–13 January 2020, 26 January 2020–28 January 2020) as fluctuations and interruptions to power supply at the site impacted the data logging system.

Second, systematic data exclusion occurred every 8 h, coinciding with the instruments filter tape progression (i.e., particulate was being collected on a new filter spot). At Liverpool, tape advances are set to occur at 0400 h, 1200 h, and 2000 h local time. Once the tape advances, there is a need for the instrument to adjust radon measurement until the tape reaches radiological equilibrium, which the manufacturer claims to take approximately 90 min (see Section 2.3.2).

In exploring the BAM radon data, we observed that after a filter change at time $t$, values at $t + 1$ h were anomalous when compared to the ANSTO measurements, particularly in the early morning when radon concentrations were near maximum (Figure 3a). This supports, to some extent, the assertion from the instrument manufacturer that it takes ~90 min for the new filter to reach radiological equilibrium. However, since our analysis is based on hourly measurements, we could not confirm the length of time required to reach equilibrium but conclude that it is >60 min and possibly up to 120 min. Consequently, we chose to exclude both the hour of and the hour following a tape change from our measurements. To ensure that data loss from the BAM due to this systematic error is minimised, we then infill the excluded data points by cubic spline interpolation (Figure 3b). For missing data at time $t$ and $t + 1$, we fit a cubic spline based on observations at $t - 2$, $t - 1$, $t + 2$, and $t + 3$, where, and only if, all those observations are valid. If not all valid, we do not interpolate. No further data filling is undertaken.

Daily, diurnal, weekly, and monthly comparisons are made using the hourly observations. As this is an exploratory analysis, we focus on simple descriptive statistics, linear regression, and coefficients of determination ($R^2$) to infer goodness of fit for linear models.

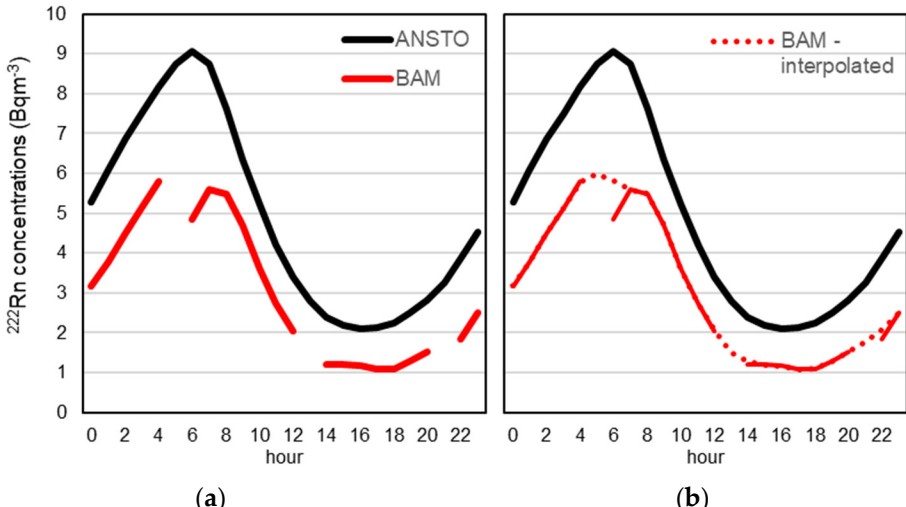

**Figure 3.** Mean diurnal profiles of $^{222}$Rn measured by ANSTO and BAM (**a**). Cubic spline interpolation of missing and low-quality BAM data used in this analysis (**b**).

## 3. Results

We first assess the annual means and distribution of the two instruments and their overall comparability. Following this is analysis of diurnal and monthly variations between the instruments and a short examination of any temperature/humidity dependencies.

### 3.1. Mean Concentrations and Distributions

Annual mean concentrations were 4.93 and 3.15 Bq/m$^3$ from the ANSTO and BAM monitors, respectively. Variance (standard deviation, σ) was 5.17 and 3.89 Bq/m$^3$, respectively. The BAM consistently measures lower concentrations than the ANSTO instrument.

Hourly observations showed a strong correlation between the instruments (R$^2$ = 0.91) and a linear relationship (Figure 4a,b). This supports the suggestion that those measurements from the BAM can be corrected to approximate the ANSTO measurements. The distributions are similar, although the BAM records significantly more lower observations (<5 Bqm$^{-3}$) and fewer high concentrations than the ANSTO instrument (Figure 4c). Nevertheless, this simple analysis supports, in the first instance, that for use cases where hourly radon observations are utilised, for example air parcel tracking and boundary layer development, the BAM may provide useful radon data to support that work.

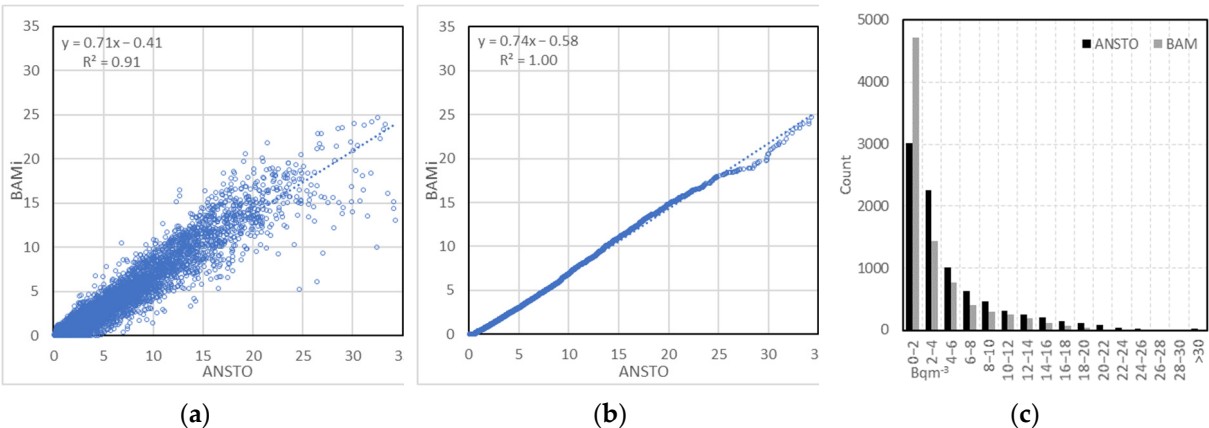

**Figure 4.** Hourly observations (Bqm$^{-3}$): scatterplot and correlation (**a**) quantile-quantile plot and correlation (**b**) and distributions (**c**).

There is similarly strong correlation between daily mean observations ($R^2 = 0.95$), and a clear linear relationship can be established (Figure 5a,b). We again conclude that a linear correction could be applied to the daily mean BAM data with some confidence. Distributions were again similar between instruments (Figure 5c). Note that the BAM recorded significantly more measurements <2 Bqm$^{-3}$, with many of these representing measurements below the minimum detection limit (MDL), and hence reported by the instrument as 0. Again, this simple assessment supports, in the first instance, that for use cases where daily radon dosages are required, for example epidemiological studies or air mass identification, the BAM data are useful.

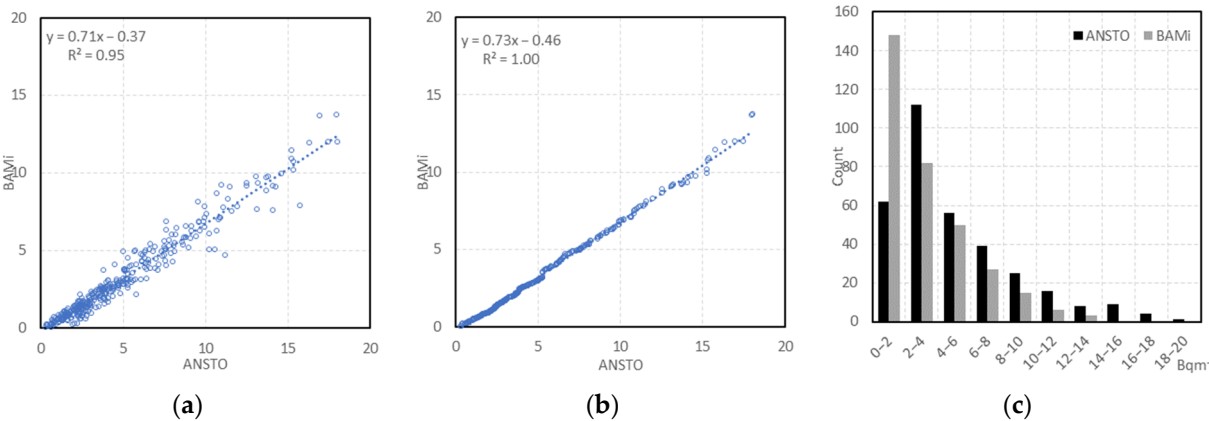

**Figure 5.** Daily mean concentrations (Bqm$^{-3}$): scatterplot and correlation (**a**), quantile-quantile plot and correlation (**b**) and distributions (**c**).

### 3.2. Diurnal, Weekday and Monthly Comparisons

Monthly and diurnal variations in radon observations are evident in both instruments (Figure 6). These variations are due to constrained boundary layer development either during the cooler months (Table 1 and Figure 7) or overnight (Figure 6a). The stable boundary layer reduces the dispersion of naturally emitted radon and leads to higher near-surface mixing ratios. During the cooler months in Sydney, there is also a much longer air mass fetch over land, resulting in a strong seasonality in monthly mean radon concentrations [13].

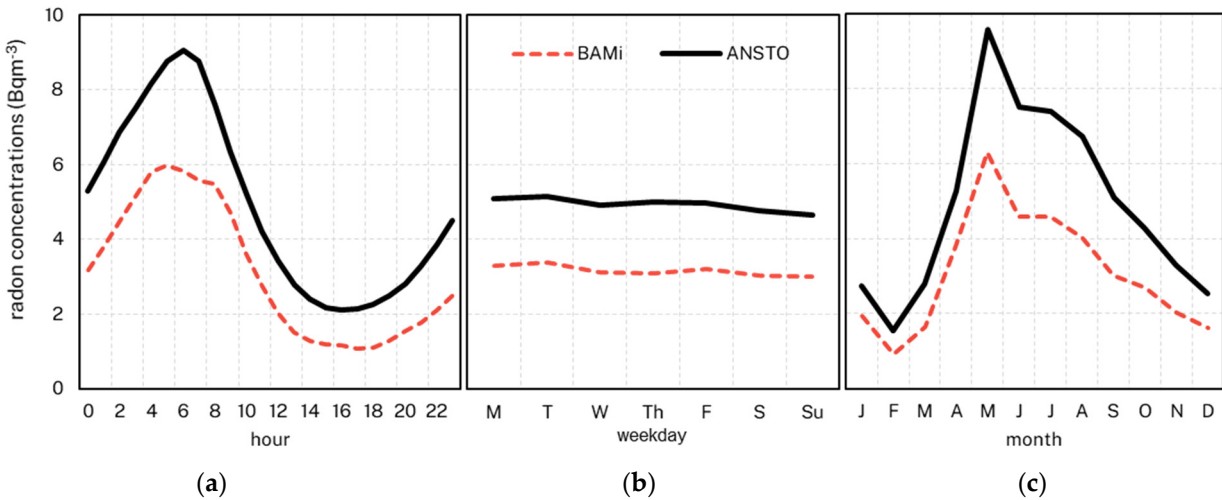

**Figure 6.** Mean radon concentrations by hour of the day (**a**), day of the week (**b**) and month (**c**).

**Table 1.** Monthly summary statistics for all observations (all values $Bqm^{-3}$).

| Period | Mean (0.95 Confidence Interval) | | σ | | Percentiles | | | | | | | | | |
|---|---|---|---|---|---|---|---|---|---|---|---|---|---|---|
| | | | | | 5th | | 25th | | Median | | 75th | | 95th | |
| | ANSTO | BAM | ANSTO | BAM | ANSTO | BAM | ANSTO | BAM | ANSTO | BAM | ANSTO | BAM | ANSTO | BAM |
| January | 2.75 (2.57–2.98) | 1.95 (1.78–2.11) | 2.68 | 2.01 | 0.50 | 0.17 | 1.05 | 0.71 | 1.90 | 1.29 | 3.35 | 2.46 | 8.09 | 6.56 |
| February | 1.56 (1.48–1.65) | 0.92 (0.83–1.00) | 1.45 | 1.11 | 0.28 | 0.00 | 0.54 | 0.16 | 1.00 | 0.52 | 2.16 | 1.28 | 4.61 | 3.55 |
| March | 2.79 (2.60–2.94) | 1.65 (1.54–1.81) | 2.54 | 1.95 | 0.37 | 0.05 | 0.90 | 0.35 | 1.92 | 0.87 | 3.74 | 2.36 | 8.26 | 6.01 |
| April | 5.26 (5.00–5.50) | 3.79 (3.52–4.06) | 4.10 | 3.47 | 0.79 | 0.09 | 1.95 | 0.90 | 4.01 | 2.63 | 7.57 | 5.81 | 13.69 | 10.89 |
| May | 9.59 (9.09–10.14) | 6.30 (5.89–6.60) | 7.23 | 5.71 | 2.02 | 0.31 | 3.79 | 1.55 | 7.29 | 4.32 | 14.01 | 10.17 | 24.06 | 17.70 |
| June | 7.50 (7.04–8.00) | 4.57 (4.27–4.93) | 6.04 | 4.64 | 1.05 | 0.00 | 2.82 | 1.08 | 5.59 | 2.84 | 10.61 | 6.80 | 20.07 | 14.87 |
| July | 7.41 (7.06–7.82) | 4.60 (4.29–4.96) | 5.80 | 4.57 | 1.38 | 0.15 | 2.92 | 1.25 | 5.57 | 3.05 | 9.99 | 6.11 | 20.20 | 14.98 |
| August | 6.72 (6.26–7.20) | 4.05 (3.75–4.40) | 6.32 | 4.63 | 0.95 | 0.05 | 2.11 | 0.71 | 4.21 | 2.05 | 9.05 | 5.45 | 20.26 | 14.94 |
| September | 5.09 (4.79–5.43) | 3.01 (2.79–3.32) | 4.92 | 3.51 | 0.71 | 0.00 | 1.79 | 0.56 | 3.15 | 1.56 | 6.56 | 4.29 | 16.84 | 11.47 |
| October | 4.29 (4.09–4.59) | 2.69 (2.48–2.93) | 3.89 | 2.99 | 0.82 | 0.00 | 1.68 | 0.66 | 2.68 | 1.63 | 5.44 | 3.46 | 13.19 | 9.95 |
| November | 3.29 (3.06–3.50) | 2.03 (1.86–2.20) | 3.23 | 2.49 | 0.68 | 0.00 | 1.42 | 0.61 | 2.16 | 1.25 | 3.65 | 2.33 | 10.96 | 7.72 |
| December | 2.52 (2.34–2.70) | 1.62 (1.46–1.76) | 2.55 | 1.87 | 0.54 | 0.00 | 0.96 | 0.41 | 1.72 | 0.98 | 2.94 | 2.09 | 8.14 | 5.93 |
| Annual | 2.75 (2.57–2.98) | 1.95 (1.78–2.11) | 5.17 | 3.89 | 0.54 | 0.00 | 1.52 | 0.60 | 2.93 | 1.58 | 6.47 | 4.16 | 16.29 | 12.04 |

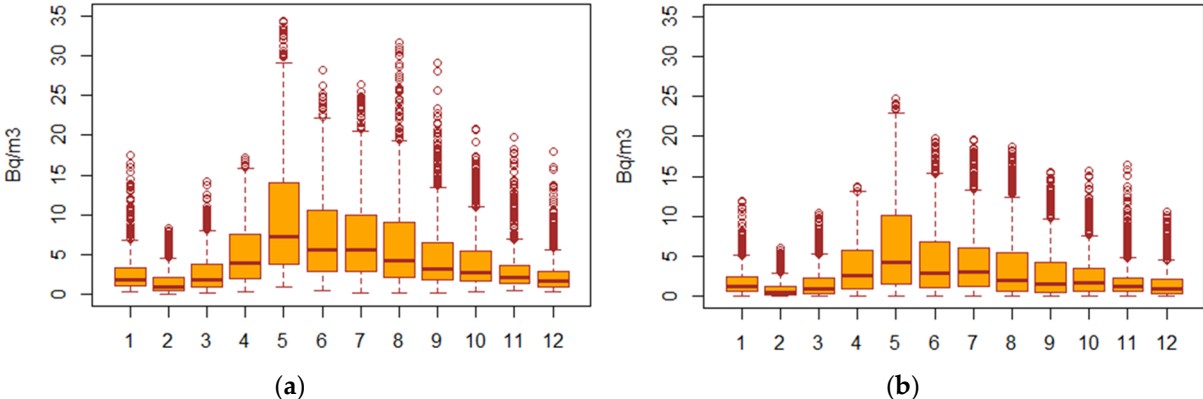

(**a**)  (**b**)

**Figure 7.** Monthly box plots for radon measured by ANSTO (**a**) and BAM (**b**).

This is illustrated clearly by investigating the diurnal profiles (Figure 6a). Radon concentrations are highest in the early morning when the stable boundary layer is most well-developed and vertical mixing is constrained. Over the course of the day, the boundary layer evolves as radiant and sensible heat transfers induce vertical mixing and the depth of the boundary layer increases [50]. As naturally occurring radon emissions on average are constant throughout the day (ignoring rainfall events and airmass advection) [51,52], observed concentrations decline in the middle of the day and late afternoon as the convective boundary layer reaches its greatest depth, dispersing radon throughout its depth.

Similarly, throughout the year we expect higher radon concentrations in the cooler months, when mean insolation is lower and convective boundary layer development is muted compared to warmer months with higher insolation (Figures 6c and 7). Both instruments clearly exhibit seasonal and diurnal variations due to boundary layer evolution. We note that maximum concentrations are observed identically in both instruments, that is between 0500–0700 h and May–August (coinciding with austral winter). The high monthly mean observations in May are likely due to the extremely dry conditions during the April–May period. Between 1 April 2019 and 3 June 2019, only 15.8 mm of rainfall was recorded at the nearby Bankstown Airport weather station, well below the long-

term average 146.9 mm for April–May. This severe rainfall deficit led to extremely dry soil conditions and likely ensured that, within radon emanating soils, air voids were maximised, leading to increased radon emissions [30,53]. May also often coincides with the transition of the subtropical ridge across the region, with more frequent anti-cyclonic conditions, large-scale subsidence, and calmer nocturnal conditions, leading to shallow stable nocturnal boundary layers with strong inversions. Higher radon concentrations in the region during May have been reported in other years [13,15].

### 3.3. Temperature, Humidity and Wind Dependence

The strong correlation between instruments, whether by hour, day, or month, suggests that there is unlikely to be much variation between the instruments induced by other environmental factors such as temperature, moisture (relative humidity), or wind. Nevertheless, we explore if there are any major deviations between the instruments due to these variables (Table 2).

**Table 2.** Correlations ($R^2$) of hourly radon observations measured by ANSTO and BAM segmented by the deciles of selected environmental variables, with 0.95 confidence intervals.

| Variable | Deciles | | | | | | | | | |
|---|---|---|---|---|---|---|---|---|---|---|
| | 1 | 2 | 3 | 4 | 5 | 6 | 7 | 8 | 9 | 10 |
| Temperature | 0.87 (0.85–0.89) | 0.89 (0.88–0.90) | 0.90 (0.89–0.91) | 0.91 (0.90–0.92) | 0.87 (0.85–0.89) | 0.87 (0.85–0.89) | 0.85 (0.83–0.87) | 0.83 (0.81–0.85) | 0.81 (0.79–0.83) | 0.76 (0.73–0.79) |
| Relative humidity | 0.68 (0.64–0.72) | 0.78 (0.75–0.81) | 0.83 (0.81–0.85) | 0.87 (0.85–0.89) | 0.89 (0.88–0.90) | 0.88 (0.86–0.90) | 0.89 (0.88–0.90) | 0.88 (0.86–0.90) | 0.94 (0.93–0.95) | 0.96 (0.95–0.97) |
| Wind speed | 0.90 (0.89–0.91) | 0.90 (0.89–0.91) | 0.89 (0.88–0.90) | 0.92 (0.91–0.93) | 0.89 (0.88–0.90) | 0.87 (0.85–0.89) | 0.84 (0.82–0.86) | 0.71 (0.68–0.74) | 0.64 (0.60–0.68) | 0.47 (0.42–0.52) |
| Sigma theta | 0.92 (0.91–0.93) | 0.89 (0.88–0.90) | 0.91 (0.90–0.92) | 0.92 (0.91–0.93) | 0.93 (0.92–0.94) | 0.91 (0.90–0.92) | 0.91 (0.90–0.92) | 0.90 (0.89–0.91) | 0.90 (0.89–0.91) | 0.89 (0.88–0.90) |

First, we consider temperature. Strong correlations ($R^2$ = 0.81–0.91) are observed for deciles 1–9, with the weakest correlation (decile 10) still substantial at $R^2$ = 0.76 (r = 0.87). This indicates that there is no impact of temperature on the relative performance of the instruments. Similarly, for relative humidity, strong correlations ($R^2$ = 0.78–0.96) are observed for deciles 2–10. Again, even the weakest correlation, decile 1 ($R^2$ = 0.68, r = 0.82), is still substantial. However, this does indicate that there may be a slightly increased variance between the instruments when conditions are very dry.

To investigate potential impacts from wind, we assess hourly vector averaged wind speed, wind direction, and sigma theta, the standard deviation of wind direction over the hour. We assess correlation between deciles for wind speed and sigma theta (Table 2) and between cardinal points for wind direction (Figure 8). There are strong correlations across all wind directions ($R^2$ = 0.82–0.93) and all deciles of sigma theta ($R^2$ = 0.89–0.93). Strong correlations were observed for wind speed deciles 1–8 ($R^2$ = 0.89–0.93) but relatively weaker correlations for higher wind speeds of deciles 9–10, where hourly vector averaged wind speeds were 3.0 ms$^{-1}$ and above. Nevertheless, even at decile 10, correlations of $R^2$ = 0.47 (r = 0.69) can be considered, at the minimum, moderate. This divergence in correlation strength at higher wind speeds may be an artifact of the separation of the monitoring stations. Even though the stations are only 2.6 km apart, in a coastal basin such as Sydney this may be sufficient separation such that the arrival of gradient flows (sea breeze, southerly change, etc.) at or near the hour may unduly impact the comparison of hourly values.

We note that there is variation in the slope of the linear regression fit between the easterly sectors (0.62–0.63) and the westerly (0.69–0.74). This may be due to the relative fetches of the air masses that predominate in each sector. Sydney is located on the coast, and westerly winds at Liverpool will more often be associated with air masses of continental origin or with extended fetch over land. In contrast, easterly winds will more frequently see air masses of marine origin or with limited fetch over land. In the easterly sector, we would expect these maritime influenced air masses to carry less radon. Consequently, there are

more hours where the BAM is returning zero as the true level (as measured by the ANSTO instrument) is below the BAMs MDL. This may be complicated by mesoscale processes such as the land-sea breeze. A complete analysis based on back trajectories and air parcel analysis/origin is beyond the scope of this paper.

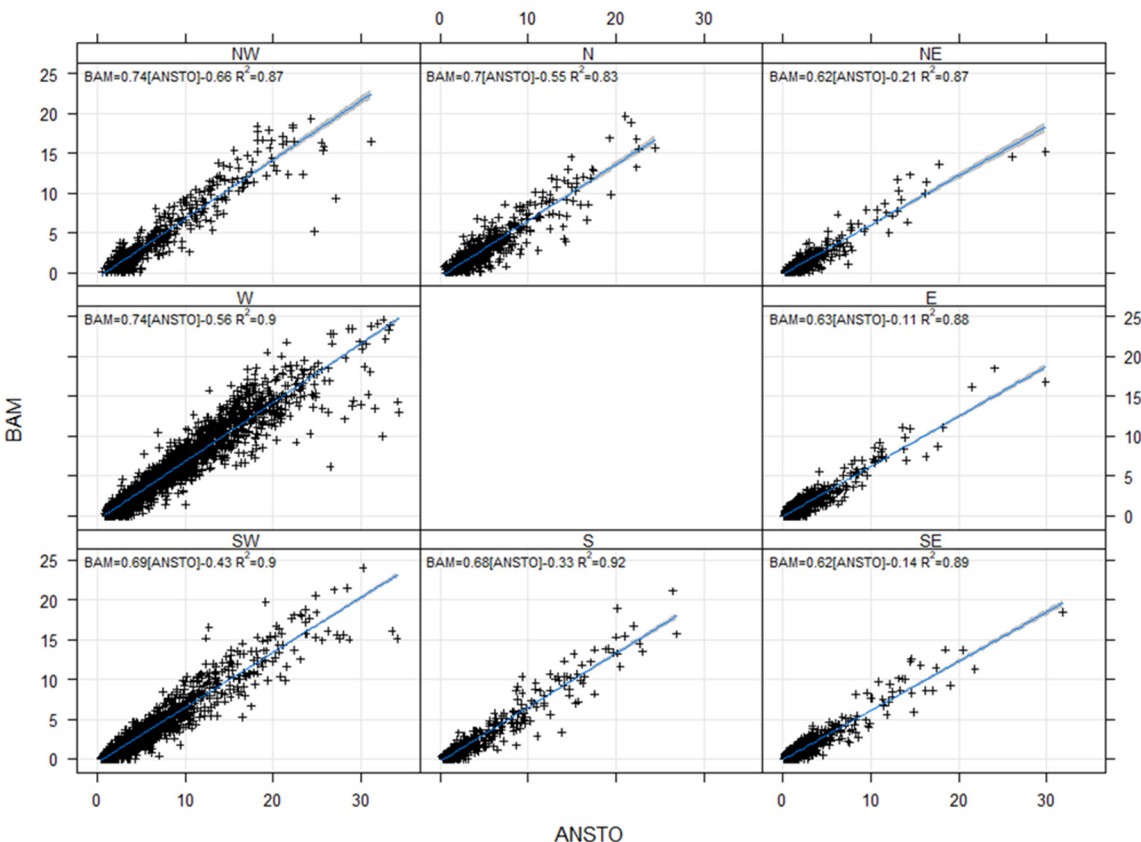

**Figure 8.** Correlations of hourly average radon measurements (Bqm$^{-3}$) by wind direction.

### 3.4. Variations under Different Atmospheric Stability Classes

Last, we explore how the performance of the instruments varies based on atmospheric stability class. Here we utilise the method of Chambers et al., (2015) [48] in order to classify atmospheric stability based solely on radon measured by the ANSTO instrument. Briefly, this approach takes single-height radon observations and conditions their time series to remove the influence of non-local processes and subsequently approximate an idealised local diurnal radon gradient. This "pseudo-gradient" is then used to derive a stability classification with four categories. Broadly these categories are: (1) near-neutral, (2) weakly stable, (3) moderately stable, and (4) stable.

Figure 9 shows the correlations and linear fit between the hourly average radon based on stability class and by season. As with the above evaluations based on temperature and wind, we see strong correlations ($R^2$ = 0.80–0.92) for conditions that are stable and moderately stable. Seasonally, correlations are strongest in winter ($R^2$ = 0.85–0.91) for all stability classes. Interestingly, even in summer we see moderately strong to strong correlations across all classes ($R^2$ = 0.72–0.90). The weakest correlations are observed for the near-neutral class in spring ($R^2$ = 0.61). The (relatively) weaker correlations for near-neutral conditions in the non-winter seasons are primarily due to the higher number of zero readings from the BAM during these seasons and conditions when actual radon (as measured by ANSTO) is below the MDL of the BAM.

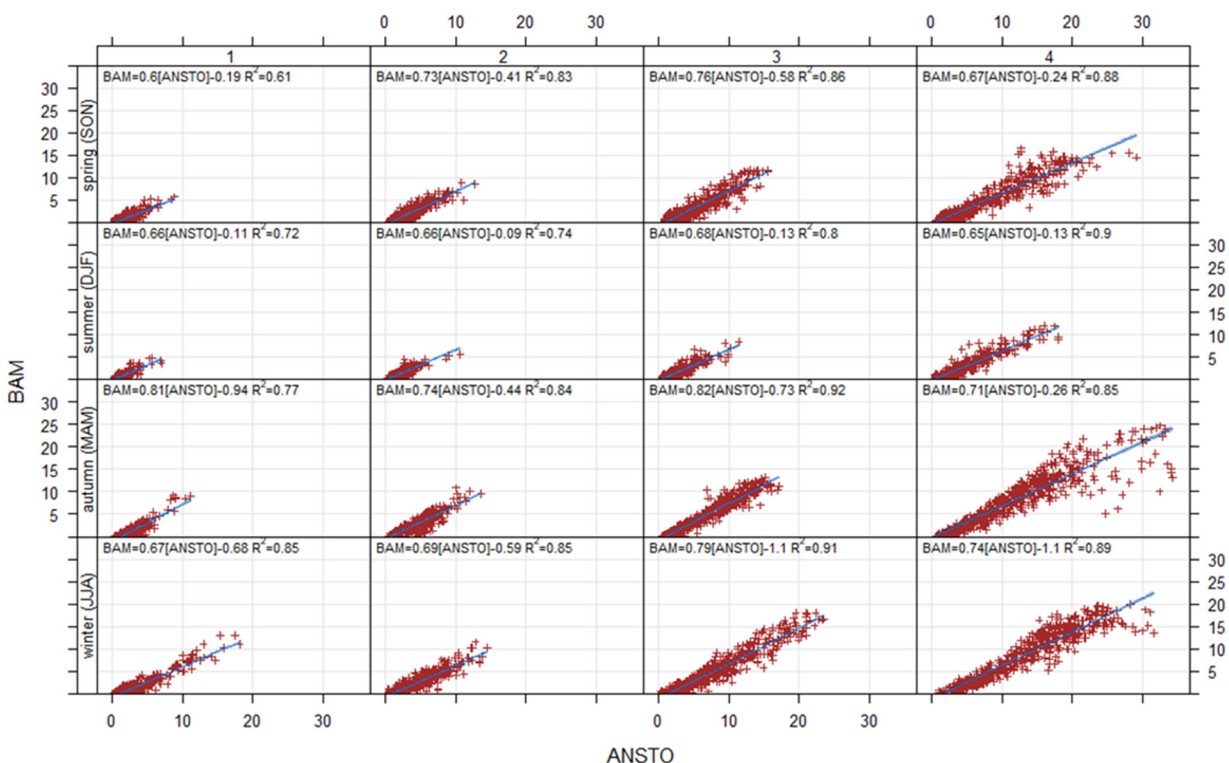

**Figure 9.** Correlations of hourly radon (Bqm$^{-3}$) by stability class and season.

## 4. Discussion

Using one year of observations at two closely sited stations, we compare radon measurements from a commercially available beta-attenuation monitor (Thermo 5014i) to the high-quality measurements from an ANSTO dual flow-loop, twin filter detector. The ANSTO detector is an integral part of the WMO's Global Atmosphere Watch program and other mature radiation monitoring networks. It has demonstrated high precision, low uncertainty, and low detection limits. The ANSTO detector acts as a de-facto global standard for radon measurement. In contrast, radon measurements in the BAM are a secondary output, being measured as a control to ensure that naturally occurring beta particles from the radon decay chain do not interfere with the measurements of the instrument's own $^{14}$C generated beta particles.

To date, we are unaware of any study that has compared radon measurements from BAMs to the ANSTO method. Although our study was for only one year and at a single site, was not conducted as a formal inter-comparison, and is limited to a small extent by the separation of the monitoring stations, we can nevertheless draw some robust conclusions.

We have demonstrated that at this site, the radon measurements from the BAM are comparable to those from the ANSTO detector. Across the 12-months there is high correlation in the hourly observations ($R^2 \sim 0.9$) and daily means ($R^2 \sim 0.95$). There is some diurnal and seasonal variation in the correlation of the hourly data. Diurnal correlations vary with strongest correlations overnight and in the early—mid-morning ($R^2 \sim 0.89–0.94$) and weakest during the afternoon (1200–1900, $R^2 \sim 0.72–0.86$). The weaker correlations during the afternoon may be a consequence of the higher MDLs of the BAM, resulting in more zero measurements during these hours. We note that this conclusion is from a site with significant coastal influence and lower radon associated with marine airmasses. At continental sites where we expect higher radon, the limitation of the MDL of the BAM may be minimised.

Monthly correlations of the hourly observations are also high, varying vary between $R^2 = 0.83$ in November and $R^2 = 0.94$ in April.

The absolute offsets observed between the BAM and the ANSTO measurements could be partially explained by the radioactive disequilibrium between $^{222}$Rn and its measured progeny close to the surface [37]. Nevertheless, in total these results suggest that the radon measurements from the 5014i BAM are strongly correlated with the ANSTO measurements.

Linear adjustments can be made to correct the BAM measurements to align to the ANSTO results. These simple linear adjustments could be applied at a range of frequencies (hourly, daily, monthly, seasonally) to best fit the purpose of the intended use of the radon measurements. In our analysis we have elected not to deal with the skewness of the observations and have performed linear regression on non-transformed data. We consider that, since the data clearly show a linear relationship, are independent, and model residuals are both homoscedastic and normally distributed, it is reasonable to fit a linear model without a log transform. This approach provides a less robust method of model fitting than if we were to perform the regression analysis and model fitting on log-log transformed data.

## 5. Recommendations

Even without adjustment, the radon measurements from BAM instruments are likely to be useful for a range of purposes. They can assist with studies where the absolute values of radon are less important than their trend or temporal response, such as when radon is used as an atmospheric tracer. This includes studies tracking airmasses, assessing boundary layer development, estimating mixing heights, verifying regional climate and chemical transport models, and inverse modelling of greenhouse gas emissions sources.

BAM instruments that measure radon internally (Thermo 5014i, Thermo 5030 SHARP, Thermo 5030i, Thermo Andersen FH62C) are widely used. In the European Union there are currently over 400 of these instruments in use (https://discomap.eea.europa.eu/App/AQViewer/index.html?fqn=Airquality_Dissem.b2g.Measurements (accessed on 1 August 2023)). While in the USA there are 79 BAM instruments with radon detection from this family reporting to the USEPA (https://www.epa.gov/outdoor-air-quality-data (accessed on 1 August 2023)). The NSW DPE network alone operates 36 of these instruments across the state of NSW, with many more spread across Australia. There are likely to be many, many more of these instruments operating globally. In contrast, currently there are only 6 sites throughout Australia where ANSTO samplers are in use.

Our study suggests that:

(1) radon measurements from the Thermo 5014i BAM are robust and precise above the MDL

(2) correlations between BAM and ANSTO measurements are strong and there is no systematic bias due to environmental variables such as temperature, humidity, wind or atmospheric stability at this site

(3) BAM radon measurements can be used "as is" for atmospheric tracer type studies, but measurements require (simple) linear adjustment, accounting for skewness, when used in studies where actual radon flux, dosage or absolute values are required

We encourage air quality monitoring network operators to log BAM radon measurements routinely, as they would for any other atmospheric variable. We further encourage operators to share BAM radon measurements freely. Similarly, researchers seeking to use radon measurements for airmass tracer type studies or for boundary layer meteorology, would benefit from establishing relationships with relevant air quality monitoring network operators. Radiation network operators should also consider the utility of BAM radon measurements for their networks.

Finally, we acknowledge that this is a single site study. We aim to expand our work to further assess the utility of radon measurements from BAMs, including assessing the response of similar BAM instruments. The NSW DPE air quality monitoring network has several monitoring stations that have around 20 years of hourly BAM radon measurements that may provide useful material for assessment of radon measurements over longer periods. We will also investigate the value of a well-designed intercomparison study at a suitable location.

**Author Contributions:** Conceptualization, M.L.R.; methodology, M.L.R. and S.D.C.; validation, M.L.R. and S.D.C.; formal analysis, M.L.R.; data curation, M.L.R. and S.D.C.; writing—original draft preparation, M.L.R. and S.D.C.; writing—review and editing, M.L.R., S.D.C. and A.G.W. All authors have read and agreed to the published version of the manuscript.

**Funding:** DPE air quality monitoring is partly funded by the NSW Climate Change Fund. ANSTO radon sampling at Liverpool occurred with funding from the National Environmental Science Program Clean Air and Urban Landscapes Hub.

**Data Availability Statement:** All data are available upon request.

**Conflicts of Interest:** The authors declare no conflict of interest.

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
