# Peer review of "Inter-Comparison of Radon Measurements from a Commercial Beta-Attenuation Monitor and ANSTO Dual Flow Loop Monitor"

_atmosphere, doi:10.3390/atmos14091333_

Round 1

Reviewer 1 Report

A very interesting work of great scientific value.

Please note the date in line 249 is missing the number 0 in the date.

Author Response

Thank you for taking the time to review our paper and your support for its publication. We have corrected the error on line 249 that you identified. 

Reviewer 2 Report

The authors address the possibility to use BAMs in order to fill gaps in radon data coverage. This type of studies could prove to be a great asses for acquiring outdoor radon data, an often missing (but valuable) parameter, used in machine learning algorithms.

Author Response

Thank you for taking the time to review our paper and your support for its publication. 

Reviewer 3 Report

All of us know the radon problem around the world and continues measurements and effective research around measuring and maps are always recommended. the history of this group in radon research is too extensive and interesting. in this study, Inter-comparison of radon measurements from a commercial beta-attenuation monitor and ANSTO dual flow loop monitor.

this paper can accept after minor change

As I see you included conclusions with recommendations also in discussion, please separate them. 

English language is fine, readable and understandable 

Author Response

Thank you for taking the time to review our paper and your support for its publication. 

We have accepted your suggestion to separate the Discussion and Conclusion sections.

Reviewer 4 Report

The manuscript shows an inter-comparison of radon measurements from a
widely used commercial beta-attenuation monitor and a state-of-art
instrument (ANSTO dual flow loop monitor). This is highly relevant
as could enable the use of data from air quality networks, widening
the spatial and temporal availability of atmospheric radon
concentration measurements. The concept is clearly described,
the manuscript is well presented and the topic is of interest to a
diverse audience. Therefore I think this manuscript is ready to be
published, except for the statistical analysis issues that I
indicate below, that should be clarified/improved before publication. i) Statistical uncertainties should be provided - Table 1, for mean and standard deviation - Table 2, for correlations ii) The data distribution is highly skewed (as shown in Fig 4c) thus
the data do not meet the assumptions in ordinary linear regression.
This is even more clear in the plots in Figures 8 and 9, with a very
high density of low concentration values. Although I don’t think this
issue affects the conclusions of the manuscript, it does influence
the quality of the fits and the corresponding estimated parameters.
Using a log-log transformation to try to reduce skewness would be a
possibility.

Author Response

Thank you for taking the time to review our manuscript. We appreciate your support and your recognition of the interest that our paper may have.

We also thank you for your suggestions regarding presentation of some of the statistics. 

As you suggested, we have added the 95% confidence intervals to the means shown in Table 1 and to the R2 values in Table 2. We have elected not to add the CIs to the standard deviations shown in Table 1, not because we reject your suggestion, but rather to ensure brevity in the table and to support a cleaner presentation of the data.

With regard to your suggestion of presenting the fitting of the linear models (for example as shown in figures 8 & 9) by adopting a log-log transformation to account for the skewness of the observations. We observe that while the data are skewed, the residuals of the fitted linear models possess normal distributions. We consider that, since the data clearly show a linear relationship, are independent, and the residuals are both homoscedastic and normally distributed, it is reasonable to fit a linear model without a log transform.

In our paper we are not trying to fit a model to correct the BAM data to ANSTO measurements, rather we are simply exploring the possibility of doing so. We agree that applying a more robust model fitting approach, such as a log-log transform, would be preferable if our aim was to transform the BAM data and demonstrate absolute comparability to the ANSTO monitor with minimum error. However, in our case we do not believe that it is necessary for our analysis given the aim of the paper. 

We have included text in the Discussion to highlight this shortcoming (following your suggestion) and to document our approach more fully. We trust that this sufficiently addresses your suggestion.